# Child and Maternal Factors Associated with Feeding Practices in Children with Poor Growth

**DOI:** 10.3390/nu15224850

**Published:** 2023-11-20

**Authors:** Dina Almaatani, Emma Cory, Julie Gardner, Mara Alexanian-Farr, Jessie M. Hulst, Robert H. J. Bandsma, Meta Van Den Heuvel

**Affiliations:** 1Department of Nutritional Sciences, Faculty of Medicine, University of Toronto, Toronto, ON M5S 1A8, Canada; dina.almaatani@mail.utoronto.ca (D.A.);; 2Department of Paediatrics, University of Toronto, Toronto, ON M5S 1A1, Canada; 3Clinical Dietetics, Hospital for Sick Children, Toronto, ON M5G 1X8, Canada; 4Division of Gastroenterology, Hepatology and Nutrition, Hospital for Sick Children, Toronto, ON M5G 1X8, Canada; 5Division of Paediatrics, Hospital for Sick Children, Toronto, ON M5G 1X8, Canada

**Keywords:** poor growth, feeding difficulty, weight concern, maternal mental health, child temperament

## Abstract

The development of adequate growth and healthy eating behaviors depends on nutritious food and responsive feeding practices. Our study examined (1) the relationship between maternal concern about child weight or perceived feeding difficulties and their feeding practices, and (2) the moderating role of child temperament and maternal mental health on their feeding practices. A cross-sessional study included mother–child dyads (n = 98) from a tertiary growth and feeding clinic. Children had a mean age of 12.7 ± 5.0 months and a mean weight-for-age z-score of −2.0 ± 1.3. Responsive and controlling feeding practices were measured with the Infant Feeding Styles Questionnaire. Spearman correlation and moderation analysis were performed. Maternal concern about child weight and perceived feeding difficulties were negatively correlated with responsive feeding (r = −0.40, −0.48, *p* < 0.001). A greater concern about child weight or perceived feeding difficulties was associated with greater use of pressure feeding practices when effortful control was low (B = 0.49, t = 2.47, *p* = 0.01; B = −0.27, *p* = 0.008). Maternal anxiety had a significant moderation effect on the relationship between feeding difficulty and pressure feeding (B = −0.04, *p* = 0.009). Higher maternal concern about child weight and perceived feeding difficulties were associated with less responsive satiety feeding beliefs and behaviors. Both child effortful control and maternal anxiety influenced the relationship between weight and feeding concerns and the use of pressure feeding practices.

## 1. Introduction

Poor growth in children is a result of the interaction between a child’s health, development, behavior, and the environment [1]. In the United States, children presenting with poor growth account for 5% to 10% of primary-care pediatric patients and 3% to 5% of pediatric hospital admissions [1]. Feeding difficulties are common in children with poor growth [2,3]. Pediatric Feeding Disorder (PFD) has been defined as a complex and heterogenous disturbance in oral intake of nutrients associated with dysfunction in the medical and nutritional status, feeding skills and safety, and/or psychosocial functioning [4]. Feeding difficulties are included in the PFD model and can be characterized as a disconnect between a caregiver (feeder) and their child and include different child feeding behaviors such as food refusal, food aversion, and limited intake of food groups [5,6].

The development of adequate growth and healthy eating behaviors depends on both nutritious food and responsive feeding practices behaviors [5,6]. Responsive feeding is a bidirectional process between a parent and a child in which the child communicates their hunger and fullness cues, and the parent responds to these cues [7]. Responsive feeding has been found to promote children’s interest in feeding, as well as their awareness of hunger and satiety cues [7].

In contrast, non-responsive feeding practices lack the reciprocity between parent and child. For example, because the caregiver can dominate or control the feeding situation [7,8]. One way in which parents employ non-responsive feeding practices is by pressuring their children to eat. Pressure to eat is a feeding practice defined by the extent to which parents use persuasion/bargaining/bribery/threats to encourage their child to eat more [9]. 

### 1.1. Parental Feeding Practices in Children with Insufficient Weight Gain and/or Poor Growth

As a response to concerns about insufficient weight gain and/or poor growth, parents can adopt controlling feeding practices which include force feeding and pressure feeding [10,11,12,13,14]. Mothers who express concerns about their underweight children have been shown to engage in pressuring feeding practices in a variety of studies in community populations around the world (e.g., Brazil [8], UK [15], the Netherlands [15], and among African American Communities in the US [13]).

Literature examining feeding practices in clinical populations of children with poor growth has been sparse. Sanders et al. examined parent–child interactions during standardized family mealtime in a clinical population of children with feeding difficulties who presented to a gastroenterology clinic [14]. The authors observed that parents used a high amount of coercive control and negative reinforcement feeding strategies in this population [14]. However, the majority of the children in this population had adequate growth (study population n = 19; 14% were at or below the 3rd percentile for weight) [14]. Further examination of feeding practices in clinical populations of children with poor growth is needed. In addition, it is unclear if child- or caregiver characteristics influence the relationship between concerns about growth/feeding difficulties and feeding practices. This could be important to the clinical management of children with poor growth.

### 1.2. Child Temperament and Parental Feeding Practices

The infant’s role in the feeding relationship is to clearly express their hunger and satiety cues and this may be influenced by their temperament [16]. Temperament has been defined as “personal characteristics that are biologically based, are evident from birth onwards, are consistent across situations and have some degree of stability” [17]. Temperament has been related to early infancy feeding difficulties and parental feeding practices [16].

A cross-sectional study of first-time Australian mothers (n = 698) showed that mothers of infants with a “negative” temperament had less awareness of their infant’s hunger cues, had a higher concern about over- and underweight, and used food to calm their children [16]. A study of preschool children in the UK (n = 241) demonstrated that a more negative temperament was associated with more emotional overeating and more feeding difficulties such as food-avoidant eating behaviors like food fussiness and less enjoyment of food [17]. Blissett et al. demonstrated that in a community population of children in the UK (n = 99) whose mothers rated them as ‘difficult’ at 1 year of age received less restrictive feeding at 2 years of age and suggested that the use of specific feeding practices may be moderated by the child’s temperament [18].

### 1.3. Maternal Mental Health and Parental Feeding Practices

A mother’s mental health is thought to influence how mothers perceive their child’s behavior, as well as how mothers interpret and respond to their child’s behavior [19,20,21]. Maternal mental health symptoms are known to interfere with the capacity to provide consistent responsive caregiving and are also related to less responsive feeding practices [22]. Depressed mothers or mothers who experience higher levels of general mental health symptoms are more likely to force feed [23,24]. In addition, mothers who report a high level of stress symptoms are also more likely to engage in forceful, controlling, and restrictive feeding practices [25,26]. A recent systematic review showed that both general and parenting stress were associated with unresponsive feeding styles [26]. Maternal anxiety has been associated with controlling and restrictive feeding practices in a longitudinal cohort study of 1-year-old infants (n = 87) in the UK [27]. Therefore, the use of specific feeding practices in children with poor growth may also be moderated by maternal mental health.

### 1.4. The Current Study

Few, and mostly older studies, have investigated child temperament and maternal mental health in clinical populations of children with poor growth [14,28,29]. However, none of these studies examined responsive feeding practices [14,28,29]. The objective of our study was to examine the correlation between maternal concern about child weight and perceived feeding difficulties and both responsive and controlling feeding practices in a clinical population of children with poor growth. Secondly, we aimed to investigate the moderating role of child temperament and maternal mental health on responsive and controlling feeding practices in this population. Identification of specific child and/or maternal factors that are associated with maternal responsive and controlling feeding practices in children with poor growth may provide an important pathway to improve the clinical management of these children. We hypothesized that child-negative temperament and maternal mental health symptoms would be correlated to non-responsive feeding practices in our study population.

## 2. Materials and Methods

### 2.1. Participants

A cross-sessional study enrolled children (6–24 months) with a first visit to the Infant and Toddler Growth and Feeding Clinic (ITGFC; Division of Pediatric Medicine) or the Nutrition Clinic (NC; Division of Gastroenterology Clinic) of the Hospital of Sick Children between April 2018 and May 2022. The Hospital of Sick Children is located in a large urban setting with a diverse ethnic population in Toronto, Canada. Both clinics are tertiary referral clinics that evaluate children with poor growth and/or feeding difficulties; the NC clinic sees mostly infants who also have gastrointestinal symptoms. Children were included if they met one of the following criteria of poor growth: (1) weight ≤ 5th percentile for age (2) weight-for-length z-score < −2 Standard Deviations (3) a weight crossing two major percentile z-scores, using WHO growth charts [30]. We excluded children with a known genetic disorder or cerebral palsy, and non-English speaking caregivers. An introduction letter about the study was sent to eligible participants and they were contacted before their clinical visits for consent. Electronic surveys (via REDCap) were emailed to consented participants who were most involved in child feeding.

This study was conducted according to the guidelines laid down in the Declaration of Helsinki and all procedures involving human subjects/patients were approved by the Hospital of Sick Children Research Ethics Board (REB) (REB number: 100005803, approval date 11 July 2017). Written informed consent was obtained from all the subjects/patients and every person received a $5 electronic gift card upon completion of the surveys.

### 2.2. Measurements

#### 2.2.1. Feeding Practices

Concern about child weight and feeding difficulties were measured with the Infant Feeding Questionnaire (IFQ) [31]. Concern about the child’s undereating or underweight was measured with 4 questions (e.g., “Do you worry that your child is not eating enough?”); reported internal consistency α = 0.71. Perceived feeding difficulties were also measured with 4 questions (e.g., “Is your child a picky eater?”); reported internal consistency α = 0.87 [31]. Questions were measured on a 5-point Likert scale, (1—never, 2—rarely, 3—sometimes, 4—often, 5—always); a higher score on the scale represents a higher level of concern about child weight and perceived feeding difficulties.

Feeding practices were measured by The Infant Feeding Styles Questionnaire (IFSQ) [32]. The IFSQ was designed to measure feeding beliefs regarding infant feeding among mothers of infants and young children [32]. In this study, we included the responsive satiety and the pressure to finish sub-scales; two feeding practices relevant to children with poor growth [14,15,32]. Responsive satiety refers to a feeding practice in which the parent is attentive to the child's hunger and satiety cues [32]. Pressure to finish refers to a feeding practice in which the parent is concerned with increasing the amount of food that the infant consumes [32]. Behavior items are scored from 1 to 5 representing response options ‘never’, ‘seldom’, ’half of the time’, ‘most of the time’, and ‘always’ [32]. Belief items are scored from 1 to 5, representing ‘disagree’, ‘slightly disagree’, ‘neutral’, ‘slightly agree’, and ‘agree’ [32]. To create a score for each of the constructs, we calculated the mean score for the items loading on that construct. A high mean score indicates a higher level of agreement with beliefs and reported practicing of behaviors. If only one item was missing in the sub-constructs, scores were calculated without that item by creating a mean score with one less item [32]. The responsive satiety domain has a total of 7 items (e.g., my child knows when she/he is full, I let the child decide how much to eat); reported internal reliability 0.92 [32]. The pressure to finish subscale included 7 items (e.g., I try to get the child to eat even if not hungry, I insist to retry a new food refusal at the same meal); reported internal reliability 0.79.

#### 2.2.2. Child Temperament

Child Temperament was measured with two instruments based on the child’s age: The Infant Behavior Questionnaire (IBQ-Very Short Form; for infants 6–12 months), internal consistency (Cronbach’s α > 0.70) [33], and the Early Behavior Questionnaire (ECBQ-VSF; for 12–36 months), internal consistency (Cronbach’s α > 0.71) [34]. Both questionnaires examined three temperament domains: (1) Extraversion/surgency, such as sociability, activity level, and positive anticipation. Children with higher surgency have typically a high activity level, show positive emotions, and enjoy intense pleasure seeking (IBQ-VSF Cronbach’s α = 0.74; ECBQ-VSF (Cronbach’s α = 0.90 for participants in our sample); (2) Negative affectivity, such as frustration, fear, and sadness. Children who have high negative affectivity show these behaviors more often without being able to be easily soothed (IBQ-VSF Cronbach’s α = 0.78; ECBQ-VSF Cronbach’s α = 0.85 for participants in our sample); and (3) Effortful control, such as attention focusing and self-regulation (IBQ-VSF Cronbach’s α = 0.64; ECBQ-VSF Cronbach’s α = 0.86 for participants in our sample). Children with higher effortful control can inhibit a dominant response (e.g., eating cookies) in order to perform a sub-dominant response (e.g., saving cookies for later) [33]. Parents were asked to rate their child on a 7-point scale ranging from 1 (extremely untrue of your child) to 7 (extremely true of your child). Higher scores indicated that children showed more of the behavior.

#### 2.2.3. Maternal Mental Health

Maternal Mental health was measured using two different questionnaires: (1) Depression Anxiety Stress Scales-21 (DASS-21); this is a 21-item self-reported questionnaire that is used for measuring depression, anxiety, and general stress in adults over one week [35]. The instrument has excellent internal consistency and a replicable three-factor structure (Cronbach’s α = 0.96) [35]. (2) Parenting Stress Index-Short Form (PSI-SF), which is a self-reported questionnaire that assesses the stress level in relation to being a parent. It is composed of 36 items and includes 3 main subscales: parental distress (PD), parent-child dysfunctional interaction (P-CDI), and difficult child (DC) [36]. The PSI includes an additional subscale “defensive responding” scale, which represents parents answering the index questions defensively either to minimize any stress or problem. Raw scores were calculated from participants’ responses and then converted to percentiles. The scores were calculated for each PSI subscale (PD, P-CDI, DC) and also for a fourth “Total Stress” subscale which combines all three subscales without the defensive responding scores. The PSI-SF has good internal consistency and reliability (Cronbach’s α = 0.91) [37].

#### 2.2.4. Sociodemographic Characteristics

Sociodemographic characteristics were collected from the child’s electronic medical record; child age and sex, birthweight (BW), gestational age (GA), and the child’s anthropometric measurements (weight-for-age z-score, height-for-age z-score, weight-for-height z-score according to WHO criteria) [38]. Mothers self-reported on maternal age, maternal education, and ethnicity and completed the Infant Toddler Checklist (ITC) [39]. The ITC is a developmental screening tool that has been validated for children between 6–24 months of age [39]. The instrument has a high degree of internal consistency (α coefficients ranging from 0.86 to 0.92) [40,41]. The ITC includes 4 composites: communication composite (e.g., Does your child do things just to get you to laugh?); Speech composite (e.g., Does your child use sounds or words to get attention or help?); Symbolic composite (e.g., Does your child show interest in playing with a variety of objects?) and a Total Composite which is the sum of the three composites [39].

### 2.3. Data Analysis

Descriptive analysis was conducted using Jamovi (The Jamovi project, version 2.3, Sydney, Australia, 2022) for baseline characteristics and reported as mean, standard deviation for continuous variables and frequencies, and percentages for categorical variables. Data were analyzed using SPSS version 23 (SPSS version 28.0 for Macintosh version, IBM Corp., Armonk, NY, 2021). Due to the fact that most variables were not normally distributed, a Spearman correlation was performed to examine the relationship between maternal concern about child weight, perceived feeding difficulties, child temperament, maternal mental health, and responsive and pressure feeding practices. Correlations below 0.30 were considered small, correlations between 0.30 and 0.50 as moderate, and correlations above 0.50 as strong [42].

Secondly, an exploratory moderation analysis was performed to evaluate if (1) specific levels of child temperament moderated the association between concern about child weight or perceived feeding difficulties and responsive and pressure feeding practices (Appendix A) and (2) if specific levels of maternal mental health moderated the association between concern about child weight or perceived feeding difficulties and responsive and pressure feeding practices (Appendix A).

The Holm-Sequential Method was used to avoid type I errors, because of the multiple comparisons. For child temperament 3 comparisons (negative affect, effortful control, surgency), and for maternal mental health 4 comparisons were included (depression, anxiety, stress, total parenting stress). First step *p*-value was calculated as *p* > 0.0125; second step *p*-value was calculated as *p* > 0.0166.

Confounding variables were identified through Spearman correlation analysis. Child age was significantly associated with surgency traits (r = 0.275, *p* = 0.007), and with depression (r = −0.216, *p* = 0.02), and anxiety (r = −0.211, *p* = 0.015). Therefore, child age was controlled in the moderation analyses.

The moderation analysis was conducted using PROCESS (Haynes, PROCESS macro; version 4.2., 2022). Where moderation effects were discovered, linear interactions were calculated by PROCESS for low, medium, and high percentiles of the moderator. Bootstrapping with 5000 samples was used to create 95% Confidence Intervals of the interaction parameter calculated in the moderation analysis, because of the non-normal distribution of some of our variables.

## 3. Results

### 3.1. Study Population

Recruitment for our study was conducted between April 2018 and May 2022, with a brief time (March 2020 to July 2020) on hold due to COVID-19 restrictions on clinical research. One hundred and forty-four (65.7%) mothers agreed to participate. Thirty-five mothers (16%) did not complete any of the questionnaires, 7 (5%) withdrew, 3 (2%) did not sign consent, and 1 (0.4%) child was excluded because of a later identified genetic disorder (Figure 1). Table 1 summarizes the child’s and mother’s sociodemographic characteristics in the study sample (n = 98). Twenty–seven (28%) mothers reported a mild to extremely severe level of depression, 31 (32%) a mild to extremely severe level of anxiety, and 45 (46%) a mild to extremely severe level of stress. As for parenting stress, n = 78 (80%) of sample mothers reported typical parenting stress percentiles and n = 4 (4%) reported clinically significant stress percentiles. Higher mean scores of responsive feeding practices (3.8 ± 0.6) were reported compared to pressure feeding practices (2.6 ± 0.6).

### 3.2. Correlation Analysis

Maternal concern about child weight and perceived feeding difficulties were both significantly and negatively correlated with responsive feeding (r = −0.40, −0.48, *p* < 0.001) but not with pressuring feeding practices (Table 2). Child negative affect was also significantly and negatively correlated with responsive feeding (r = −0.24, *p* = 0.02) and not with pressure feeding practices. As for maternal mental health, both depression and all domains of parenting stress were significantly and negatively correlated to responsive feeding practices (r = −0.28, *p* = 0.009; Total Parenting Stress r = −0.37, *p* < 0.001, Table 2). Only maternal anxiety and perceiving a child as difficult on the parenting stress scale were significantly and positively correlated with pressure feeding practices (r = 0.34, *p* = 0.002; r = 0.32 *p* = 0.003; Table 2).

### 3.3. Moderation Effect of Child Temperament

All three temperament domains did not have a moderate effect on the relationship between concern for child weight or perceived feeding difficulties and responsive feeding practices (Table 3). Effortful control had a significant moderation effect on the relationship between concern about child weight and pressure feeding (R = 0.42, R-sq = 0.1779, B = −0.36, *p* = 0.001; Figure 2A). A greater concern about child weight was associated with greater use of pressure feeding practices when effortful control was low (R = 0.42, R-sq = 0.1779, B = 0.49, t = 4.03, *p* = 0.001). A similar effect of effortful was observed on the relationship between perceived feeding difficulties and pressure feeding (R = 0.38, R-sq = 0.148, B = −0.27, *p* = 0.008; Figure 2B). Higher perceived feeding difficulties were associated with more use of pressure feeding when effortful control was low (R = 0.38, R-sq = 0.148, B = 0.47, t = 3.55, *p* = 0.0007). Both negative affect and surgency did not have a significant moderation effect on the relationship between concern for child weight or perceived feeding difficulties and pressure feeding practices.

### 3.4. Moderation Effect of Maternal Mental Health

Maternal depression, general stress, or parenting stress had no significant moderation effect on the relationship between maternal concern about child weight and feeding difficulty and both responsive and pressure feeding practices (Table 4).

Only maternal anxiety had a statistically significant moderation effect on the relationship between feeding difficulty and pressure feeding (R = 0.42, R-sq = 0.177, B = −0.04, *p* = 0.009). When there were lower perceived feeding difficulties, high anxiety was associated with higher use of pressure feeding (R = 0.42, R-sq = 0.177, B = 0.27, t = 2.43, *p* = 0.017; Figure 3B). However, this moderation effect was not significant in children with higher perceived feeding difficulties (R = 0.42, R-sq = 0.177, B = −0.25, t = −1.67, *p* = 0.09; Figure 3). Maternal anxiety had a similar moderation effect on the relationship between maternal weight concern and pressure feeding (B = −0.037, *p* = 0.011; Figure 3A).

## 4. Discussion

This study examined the relationship between maternal concern about child weight and perceived feeding difficulties and responsive and pressure feeding practices. Although this relationship has previously been examined in different community cohorts [8,11,12,24,43,44], our results are specific to children with poor growth in a clinical population. High concerns about child weight and/or perceived feeding difficulties were moderately correlated with less awareness of responsive satiety cues but not with the use of pressure feeding.

In line with our hypothesis, we found that higher negative child temperament was negatively correlated with responsive feeding practices. If caregivers rated their child as having higher frustrated behavior and as being more difficult to soothe, they also described being less responsive to the child’s hunger and satiety cues; as previously reported in community populations [16]. Maternal depression and all domains of parenting stress showed a small negative correlation with responsive satiety feeding practices in our study. Mothers who experience depression may also be less responsive to hunger and satiety cues [23,24,45]. There is a paucity of literature that examined the association between parenting stress and responsive feeding practices in children [26]. We identified only one study by Hughes et al. that reported that parenting stress was significantly associated with an uninvolved feeding style in families with a low income [46]. As responsive feeding is related to responsive parenting, addressing parenting stress may provide a pathway to promote responsive feeding practices in children presenting with poor growth [7].

However, unlike reports from community populations, our study did not identify a significant correlation between concern about child weight or perceived feeding difficulties and pressure feeding. This finding contrasts with a recent systematic review and meta-analysis that examined the relationship between concern about child weight and non-responsive feeding practices among parents of healthy children in the community aged 1–11 years [8]. In this systematic review, parental concern about child underweight was a significant risk factor for using pressure feeding practices [8]. The studies in the meta-analysis included only two community populations of mother and child dyads in Brazil of older children (2–8 years old). The demands on mothers in a clinical setting are likely much higher than in a community population so we were surprised by the lack of correlation between weight and feeding concerns and pressure feeding. A possible explanation could be that in our study population, both child temperament and maternal mental health factors were associated with the use of pressure-feeding practices.

A novel finding was the association between child effort and the use of pressure-feeding practices in our study. The moderation analysis showed that mothers with greater concern about weight or perceived feeding difficulties used significantly more pressure-feeding practices in children with low effortful control. Low effortful control is related to low self-regulation and low inhibitory control; children with low effortful control may have difficulties paying attention and “staying on task” [47]. Caregivers of children with poor growth may use pressure feeding practices in children with low effortful control to increase their child’s attention to the “task” of eating.

Maternal anxiety and rating the child as difficult on the PSI were also moderately correlated with the use of pressure feeding practices. This indicates that mothers who experience anxiety or rate their children as difficult may use more pressuring feeding practices. Limited literature has investigated the role of maternal anxiety and feeding practices in community populations. One study by Blissett et al. described that maternal anxiety was positively correlated with the use of restrictive feeding practices but not with pressure feeding practices in a population of 1-year-old infants [27]. Our moderation analysis showed that mothers with high anxiety symptoms and who had low concerns about perceived feeding difficulties used more pressure feeding practices. However, the moderating role of maternal anxiety was not significant in mothers who perceived a high amount of feeding difficulties. This finding shows that having high concerns about feeding difficulties is associated with the use of pressure-feeding practices independent of maternal anxiety in our population of children with poor growth.

Strengths of our study include its focus on responsive and pressure-feeding practices with the use of validated measures in an ethnically diverse clinical population of children with poor growth. However, our study limitations should also be acknowledged. Although we were able to collect many sociodemographic variables, we did not collect maternal nativity, employment, or marital status. Our study results need to be interpreted with caution because of possible selection bias. Our study relies on self-report and 67.5% of mothers agreed to participate in our study. Mothers with children with poor growth spend a lot of time feeding their children. It could be that mothers who had the most concerns about child weight and or feeding difficulties did not participate in our study, for example, because they did not find the time to answer the questionnaires; our study results may have therefore been underestimated. Secondly, there could have been a response bias; mothers might have felt pressure to give socially acceptable answers. Children were seen in a tertiary referral clinic and our mothers were highly educated, mothers were likely given information on responsive feeding practices before referral to our clinic. In the PSI (13.5%) of sample mothers responded defensively, making it likely that other questionnaires were defensively answered. Although we included maternal psychopathology in our study, we did not inquire about eating disorder psychopathology which could also influence feeding practices [48]. In addition, because of our limited sample size, we could not adjust for all confounding sociodemographic factors (including household income) that may influence feeding practices. Finally, because of the cross-sectional study design no causal relation could be determined.

## 5. Conclusions

Higher maternal concern about child weight and perceived feeding difficulties were associated with less responsive satiety feeding beliefs and behaviors. This may lead to less responsive feeding practices in which the caregiver ignores the hunger and satiety cues of the child. Our results emphasize the importance of addressing maternal concerns about child weight and perceived feeding difficulties in the clinical assessment of children with poor growth to improve responsive feeding. Both child self-regulation and maternal anxiety influenced the relationship between weight and feeding concerns and the use of pressure-feeding practices. In order to improve responsive feeding in a clinical population of children with poor growth practitioners are encouraged to take a child’s temperament and maternal mental health into account, with a more tailored approach to address maternal concerns about weight and perceived feeding difficulties.

## Figures and Tables

**Figure 1 nutrients-15-04850-f001:**
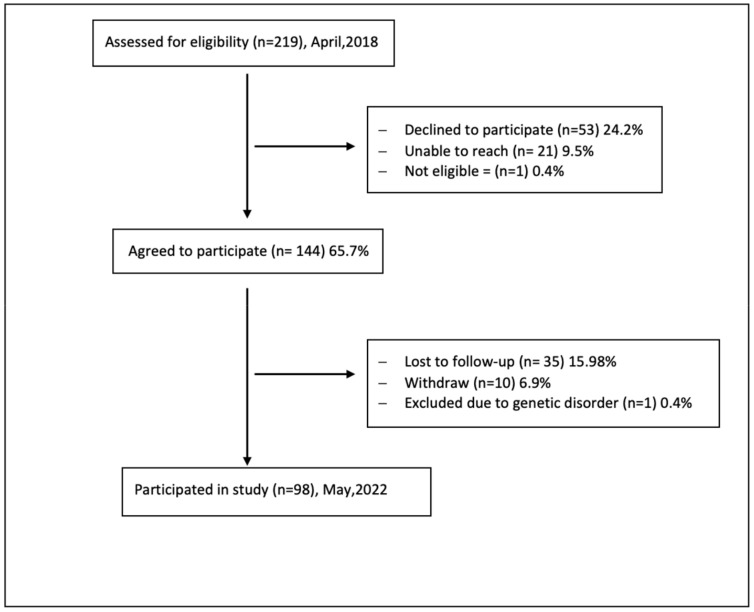
Recruitment flow diagram.

**Figure 2 nutrients-15-04850-f002:**
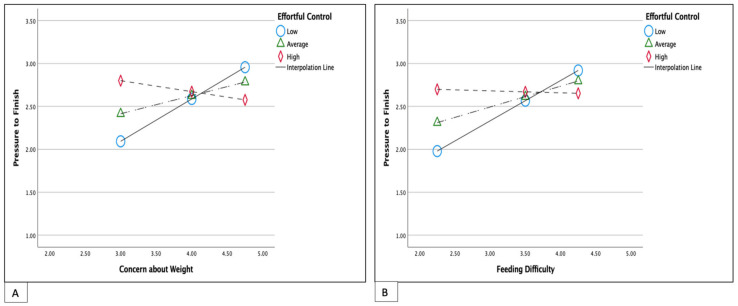
Simple slope equations of the relationship between maternal concern about child weight (**A**) or feeding difficulties (**B**) and pressure feeding practices when children’s effortful control is low, average, or high. Low effortful control 4.0; Average effortful control 4.8; High effortful control 5.8. Both models adjusted for child age.

**Figure 3 nutrients-15-04850-f003:**
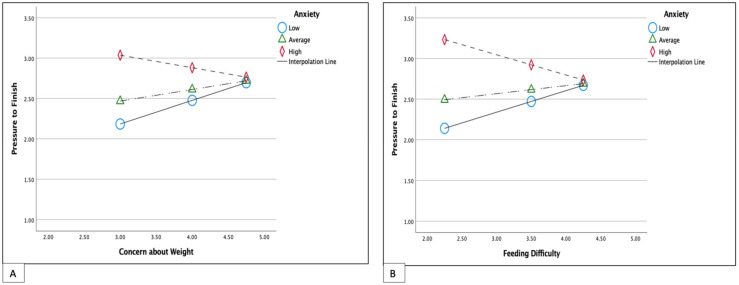
Simple slopes equations of the relationship between maternal concern about child weight (**A**) and feeding difficulty (**B**) and pressure feeding when maternal anxiety is low, average, or high. Low anxiety; 0.0; Average anxiety 4.0, High Anxiety 12.4. Both models adjusted for child age.

**Table 1 nutrients-15-04850-t001:** Sociodemographic Characteristics of the Study Population.

Child Characteristics	N = 98
Age in months (mean ± standard deviation)	12.7 ± 5.03
Sex (male, percentage)	50 (51.5)
Weight-for-age z-scores (mean ± standard deviation)	−2.01 ± 1.31
Height-for-age z-scores (mean ± standard deviation)	−1.54 ± 1.57
Weight-for-height z-scores (mean ± standard deviation)	−1.49 ± 1.14
Birth weight (kg, mean ± standard deviation) (=71)	3.22 ± 4.26
Gastro-esophageal reflux (cases, percentage) (=87)	23 (26.4)
*Gestational age* (cases, %) (n = 91)	
Term	66 (72.5)
Premature	25 (27.5)
Small for gestational age (cases, %) (n = 79)	34 (44.7)
*Child Development* [39] (concern, n, %)	
Communication composites	21 (22.3)
Speech composites	17 (18.1)
Symbolic composites	25 (26.6)
Total composites	22 (23.4)
*Child Temperament*^a^ [33,34] (mean ± standard deviation)	
Surgency	5.01 ± 1.13
Negative affectivity	3.68 ± 1.24
Effortful control	4.82 ± 1.1
**Maternal characteristics**	
Age (years) (mean ± standard deviation)	35.1 ± 5.88
*Ethnicity* (cases, percentage)	
Western European	17 (17.3)
South Asian	31 (31.6)
Eastern European	8 (8.16)
East Asian (Chinese)	8 (8.16)
Caribbean Region	4 (4.08)
Southeast Asian	8 (8.2)
West Asian	4 (4.08)
Other	18 (18.36)
*Education* (cases, percentage)	
University degree	69 (70.4)
No certificate or degree	2 (2)
High school or equivalent	9 (9.2)
College or non-university certificate	16 (16.3)
Apprenticeship or trades certificate or diploma	2 (2)
*The Depression Anxiety Stress Scale* [35] (cases, percentage) (n = 97)	
Depression	
Normal	70 (72.2)
Mild–Moderate	20 (20.6)
Severe–Extremely severe	7(7.2)
*Anxiety*	
Normal	66 (68.0)
Mild–Moderate	21(21.6)
Severe–Extremely severe	10 (10.3)
*Stress*	
Normal	52 (53.6)
Mild–Moderate	39 (40.2)
Severe–Extremely severe	6 (6.2)
*Parenting Stress Index-Short Form*^b^ [36] (cases, percentage) (n = 96)	
*Total stress*	
Typical Stress Percentiles ^c^	78 (81.3)
High Stress Percentiles ^d^	1 (1)
Clinically Significant Stress Percentiles ^e^	4 (4.2)
Defensive response ^f^	13 (13.5)
*Parental Distress (PD)*	
Typical Stress Percentiles	70 (72.9)
High Stress Percentiles	6 (6.3)
Clinically Significant Stress Percentiles	11 (11.5)
*Parent-Child Dysfunctional Interaction (P-CDI)*	
Typical Stress Percentiles	72 (75)
High Stress Percentiles	2 (2.1)
Clinically Significant Stress Percentiles	4 (4.2)
*Difficult Child (DC)*	
Typical Stress Percentiles	67 (69.8)
High Stress Percentiles	2 (2.1)
Clinically Significant Stress Percentiles	7 (7.3)

^a^ Child Temperament score range: Surgency 2.3 ≤ 6.8, Negative affectivity 1.8 ≤ 6.4, Effortful control 2.9 ≤ 6.7. ^b^ Percentile scores describe parent’s relative standing among all the parents who were assessed during the development and testing of the PSI instrument. ^c^ Typical stress percentile score (15–80). ^d^ High stress percentile score = P-CDI (81–84) and (81–89) other sub-scales. ^e^ Clinically significant stress percentiles score = P-CDI (85–100), and (90–100) for other sub-scales. ^f^ Defense responsive percentile score (<10).

**Table 2 nutrients-15-04850-t002:** Correlation between Concern about Child Weight, Feeding Difficulty, Child Temperament and Maternal Mental Health and Responsive and Pressure Feeding Practices.

	Responsive Satiety		Pressure to Finish	
	Correlation Coefficient	*p*-Value	Correlation Coefficient	*p*-Value
*Feeding practice beliefs*				
Concern about a child's weight	−0.40	<0.001	0.18	0.15
Feeding Difficulty	−0.48	<0.001	0.21	0.09
*Child Temperament*				
Negative Affect	−0.24	0.04	0.11	0.26
Surgency	0.08	0.38	0.07	0.52
Effortful Control	0.14	0.19	0.08	0.72
*Maternal Mental Health*				
Depression	−0.28	0.02	0.05	0.98
Anxiety	−0.19	0.10	0.34	0.006
Stress	−0.19	0.10	0.12	0.50
Parenting Stress (total)	−0.37	0.001	0.18	0.20
Parental Distress	−0.24	0.04	−0.07	0.31
Parent-child dysfunction	−0.29	0.01	0.16	0.19
Difficult Child score	−0.33	0.004	0.32	0.007

**Table 3 nutrients-15-04850-t003:** Moderation Analyses Exploring the Role of Temperament in the Relationship between Mothers’ Perceptions about Child Weight and Feeding Difficulty and Their Feeding Practices.

Feeding Perception	Feeding Practices	Child Temperament	B	95%CI ^1^	*p*
Concern Child Weight	Responsive Satiety	Negative Affect	−0.04	−0.16; 0.10	0.52
		Effortful Control	−0.12	−0.32; 0.07	0.22
		Surgency	−0.09	−0.35; 0.11	0.30
Concern Child Weight	Pressure to finish	Negative Affect	−0.12	−0.30; 0.07	0.14
		Effortful Control	−0.36	−0.56; −0.12	0.001
		Surgency	0.10	−0.09; 0.35	0.31
Feeding Difficulty	Responsive Satiety	Negative Affect	−0.14	−0.25; −0.01	0.06
		Effortful Control	−0.02	−0.21; 0.17	0.78
		Surgency	−0.19	−0.40; −0.01	0.02 ^2^
Feeding Difficulty	Pressure to finish	Negative Affect	−0.10	−0.27; 0.07	0.23
		Effortful Control	−0.27	−0.51; −0.07	0.008
		Surgency	0.11	−0.10; 0.34	0.20

^1^ 95% Confidence Interval (CI) obtained with Bootstrapping with 5000 bootstrap samples. ^2^ The *p*-value of surgency was 0.0153; this was bigger than the *p*-value determined with the Holms-Sequential Method which we calculated (*p* = 0.0125).

**Table 4 nutrients-15-04850-t004:** Moderation Analyses Exploring the Role of Maternal Mental Health in the Relationship between Maternal Perceptions about Child Weight and Feeding Difficulty and their Feeding Practices.

Feeding Perception	Feeding Practices	Maternal Mental Health	B	95% CI ^1^	*p*
Concern Child Weight	Responsive Satiety	Depression	0.02	0.00; 0.04	0.04 ^2^
		Anxiety	0.01	−0.01; 0.04	0.21
		Stress	0.01	−0.00; 0.03	0.11
		Total Parenting Stress	0.00	−0.03; 0.03	0.50
Concern Child Weight	Pressure to finish	Depression	−0.02	−0.05; 0.01	0.08
		Anxiety	−0.04	−0.07; −0.01	0.01
		Stress	−0.00	−0.03; 0.02	0.74
		Total Parenting Stress	−0.00	−0.02; 0.01	0.31
Feeding Difficulty	Responsive Satiety	Depression	0.00	−0.03; 0.02	0.88
		Anxiety	0.00	−0.02; 0.03	0.82
		Stress	0.01	−0.01; 0.02	0.35
		Total Parenting Stress	−0.00	−0.01; 0.00	0.63
Feeding Difficulty	Pressure to finish	Depression	−0.01	−0.04; 0.03	0.57
		Anxiety	−0.04	−0.08; −0.01	0.01
		Stress	0.00	−0.02; 0.03	0.90
		Total Parenting Stress	0.00	−0.01; 0.01	0.89

^1^ 95% Confidence Interval (CI) obtained with Bootstrapping with 5000 bootstrap samples. ^2^ The *p*-value of depression was 0.04; this was bigger than the *p*-value determined with the Holms-Sequential Method which we calculated (*p* = 0.0125).

## Data Availability

Data are available upon request.

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
