# Peer review of "Child and Maternal Factors Associated with Feeding Practices in Children with Poor Growth"

_nutrients, 2023, doi:10.3390/nu15224850_

Round 1

Reviewer 1 Report

Comments and Suggestions for Authors

Are there any other factors that could influence your study such as economic factors of the family? 

CHildren with poor growth usually come from families without the adequate family infrastructure to help with the new born or have economic issues or disparities.

Author Response

Are there any other factors that could influence your study such as economic factors of the family? 

We thank the reviewer for this feedback. We agree, economic factors, such as household income could influence the results of our study. Unfortunately, we could not adjust for household income in our analysis because of our small sample size. We have made this limitation more clear in the Discussion section; line 408

Children with poor growth usually come from families without the adequate family infrastructure to help with the new born or have economic issues or disparities.

See our feedback above.

Reviewer 2 Report

Comments and Suggestions for Authors

This is a generally well-written manuscript assressing an important topic, fitting the aims of the journal and potentially adding to previous literature.

I have some suggestions that could improve the paper.

The introduction section is well-structured and conveys almost all the information useful to the reader to understand the framework of the research. However, one possible limit is that no reference is done with regards to possibile differences between mothers and fathers' feeding styles. There is a - not so small - amout of literature that addressed this point, in particular about pressure to finish feeding and control practices in mothers and fathers. I suggest adding at least a brief passage about this point.

If possible, I think it could be useful adding specific aims and hypotheses (that are currently present in a general form at the end of the introduction section). These specific aims should then match the order in which the results are presented and the discussion in organized.

Although the influence of individual psychopathological problems in parents is acknowledged by the authors as possibly associated with the other study variables, it seems to me that no reference is present with regards to parental problems in the area of eating (disordered eating and/or eating disorders). There is a large bulk of literature about how parents' difficulty in such area can influence their feeding practices with their own offspring. I think the authors should consider the previous literature on this point and possibly adding this question as a limitation.

Another point to consider is the potential influence of genetic/epigenetic characteristics of children. Again, there is literature on this and i suggest adding at least a passage about this point.

The method section is clear and well detailed.

The discussion is correctly discuss the results.

Comments on the Quality of English Language

The quality of english language is fine, with some minor mistakes an typos.

Author Response

This is a generally well-written manuscript assessing an important topic, fitting the aims of the journal and potentially adding to previous literature.

 We thank the reviewer for this positive feedback.

I have some suggestions that could improve the paper.

The introduction section is well-structured and conveys almost all the information useful to the reader to understand the framework of the research. However, one possible limit is that no reference is done with regards to possibile differences between mothers and fathers' feeding styles. There is a - not so small - amout of literature that addressed this point, in particular about pressure to finish feeding and control practices in mothers and fathers. I suggest adding at least a brief passage about this point.

 We thank the reviewer for this feedback. Although we agree with the reviewer that investigating the feeding practices of fathers is very important, our paper focuses on "maternal factors" associated with feeding practices. Fathers did not fill out any of the questionnaires. We therefore feel this is a separate topic and have not addressed it in our manuscript.

If possible, I think it could be useful adding specific aims and hypotheses (that are currently present in a general form at the end of the introduction section). These specific aims should then match the order in which the results are presented and the discussion in organized.

 In the paragraph "The current study" we have included our study objective and secondary aim ; The objective of our study was to examine the correlation between maternal concern about child weight and perceived feeding difficulties and both responsive and controlling feeding practices in a clinical population of children with poor growth. Secondly, we aimed to investigate the moderating role of child temperament and maternal mental health on responsive and controlling feeding practices in this population. Both the result section and discussion are organized to discuss these objectives in this order.

Although the influence of individual psychopathological problems in parents is acknowledged by the authors as possibly associated with the other study variables, it seems to me that no reference is present with regards to parental problems in the area of eating (disordered eating and/or eating disorders). There is a large bulk of literature about how parents' difficulty in such area can influence their feeding practices with their own offspring. I think the authors should consider the previous literature on this point and possibly adding this question as a limitation.

 We agree with the reviewer that parent's eating-disorder psychopathology was not accounted for in this study. We have added this to our limitation section. Line 407

Another point to consider is the potential influence of genetic/epigenetic characteristics of children. Again, there is literature on this and i suggest adding at least a passage about this point.

We think this is out of the scope of our study; although genetics due influence children's body weight  and subsequent feeding practices; we have not identified literature that described a direct genetic influence on feeding practices.

The method section is clear and well detailed.

We thank the reviewer for this feedback.

The discussion is correctly discuss the results.

We thank the reviewer for this feedback.

Reviewer 3 Report

Comments and Suggestions for Authors

An interesting topic that certainly is of value to the clinical community.

I think the use of 'feeding difficulties' should be included in a wider definition of Paediatric Feeding Disorder, so it's place with other difficulties can be determined.

The definition of 'pressure to feed' includes the word pressure. It would be more useful to characterise 'pressure' as persuasion/ bargaining/ bribery/ threats.

Line 56 and 57 - there are too many 'practices' in this one sentence - could be reworded to expand meaning. This paragraph feels like a repeat of the previous paragraph and the two could be combined, then expanded upon.

It is important to clarify this is about 'insufficient' weight gain, as too much weight gain can also be problematic and is not the focus of this study. Likewise these are parental feeding practices, not those of the child.

The section on childhood temperament might be better described in terms of ages and stages of development as currently it shifts back and forth between infants then preschoolers, then toddlers.

It would be useful to state the broad demographic of the city, and the country as the context will be relevant to parenting.

Line 152 - typo 'beliefs'

Line 161  - the constructs or just construct

line 168 - the second example doesn't make sense

The methods section is dense and could benefit from some subheadings. Data analysis would benefit from being broken into smaller paragraphs.

line 229-230 sentence starting with 'because' needs fixing. In general, it might flow better to start sentences with words other than 'because'. At times in this section you use the third person as well as the first person, using just one perspective would help the flow. I suggest the first person 'we' as it is easier to read.

Given that the participants are in the study after being referred to a tertiary service, it would be important to know what previous input they have received. If they have had advice from health professionals they may have been discouraged from pressure feeding, and already be aware of some of the recommended practices for their child. This would be a significant difference between this study population and the general population that you are comparing to in other studies. Also how does your sample of educated mothers compare with other study samples, what influence does education have on their ability to seek information and help?

line 382 does not need a hyphen.

In the conclusion, a more practical, or behavioural description of 'effortful control' would help the reader apply the findings.

Comments on the Quality of English Language

English language is fine with only minor editting needed to correct some incomplete sentences and aspect changes.

Author Response

See attached feedback.
